# The Presence of PDL-1 on CD8+ Lymphocytes Is Linked to Survival in Neonatal Sepsis

**DOI:** 10.3390/children9081171

**Published:** 2022-08-04

**Authors:** Lyudmila L. Akhmaltdinova, Zhibek A. Zhumadilova, Svetlana I. Kolesnichenko, Alyona V. Lavrinenko, Irina A. Kadyrova, Olga V. Avdienko, Lyudmila G. Panibratec, Elena V. Vinogradskaya

**Affiliations:** 1Shared Resource Laboratory, Karaganda Medical University, Karaganda 100000, Kazakhstan; 2National Scientific Cardiac Surgery Center, Nur-Sultan 010000, Kazakhstan; 3Regional Perinatal Center, Karaganda 100000, Kazakhstan

**Keywords:** neonatal sepsis, neonates, infants, CD8+ T cells, laboratory marker, programmed death receptor, PD-1

## Abstract

Sepsis is life-threatening organ dysfunction caused by a dysregulated host response to infection. Neonatal sepsis is the main cause of death in newborns, especially preterm infants. The pathogenesis of sepsis is based on a hyper-inflammatory syndrome combined with an immunosuppressive mechanism in sepsis. This study aimed to find critical parameters that are associated with the outcome of newborns with suspected sepsis. Understanding the association might have clinical relevance for immuno-monitoring, outcome prediction, and targeted therapy. Methods: A total of 210 newborn infants no older than 4 days with suspected sepsis at admission in Karaganda (Kazakhstan) were prospectively enrolled. Blood cultures were incubated, and pathogens in positive cultures were determined by MALDI-TOF. An immunological assay for blood cell components was conducted by flow cytometry with antibody cocktails. The diagnostic criteria for neonatal sepsis were identified by qualified neonatologists and included both clinical sepsis and/or positive blood culture. The analyzed infants were grouped into non-septic infants, surviving septic infants, and deceased septic infants. The results showed that deceased septic newborns had a lower level of CD8+ lymphocytes and higher PDL-1 expression in comparison with surviving septic newborns. PDL-1 expression on CD8+ T cells might play an immunosuppressive role during neonatal sepsis and might be used as a laboratory biomarker in the future.

## 1. Introduction

Sepsis is one of the primary neonatal diseases in terms of the frequency and severity of outcomes. The frequency of sepsis in the pediatric population is increasing, mainly due to the higher survival rate of very low birth weight infants and children with chronic diseases [1]. Neonatal sepsis remains a leading cause of infant mortality in Kazakhstan and accounts for up to 25.9% of all infant deaths. At the same time, infant mortality in the Karaganda region was 11.9 per 10,000 live births in 2020, and the recent years of the pandemic have led to increased mortality [2].

Nowadays, the diagnosis of neonatal sepsis is still based on the criteria for systemic inflammatory response syndrome (SIRS) [3]. This approach to sepsis recognition is highly sensitive but lacks specificity. Traditional laboratory biomarkers that are used in neonatal sepsis, such as C-reactive protein (CRP) and procalcitonin tests, are not effective alone as laboratory markers. Patients with sepsis and septic shock are heterogeneous groups with different underlying pathophysiological mechanisms [3,4]. Moreover, this heterogeneity is most pronounced in newborns due to adaptive mechanisms in the first days and weeks of life.

The neonate’s innate immune system is underdeveloped, which increases the risk of sepsis development. In particular, the innate immune response to infection begins with neutrophils, but infants have physiological neutropenia, and neutrophils have inhibited migratory and phagocytic abilities [5]. Monocytes, which produce antimicrobial components such as CRP, have diminished antigen-presenting abilities in newborns [6]. Furthermore, neonates are unable to initiate a memory response because acquired immunity is deficient in the neonate. These factors lead to increase vulnerability to infection during the neonatal period.

Previously, sepsis has been attributed to the manifestation of a hyper-inflammatory syndrome, but recent data have shown that the mechanism of systemic inflammation is more complicated. There is growing evidence to support the role of immunosuppression in sepsis [7]. However, its role in neonatal sepsis has not yet been determined. One of the key models in the development of sepsis-mediated immunosuppression is programmed death receptor (PD) PD-1/PDL-1, but its role in newborns is still poorly described [5,7]. PD-1 plays an important role in the immune response by down-regulating T cell activity during immune responses to prevent autoimmune tissue damage.

PD-1/PDL-1 exerts its inhibitory effect by limiting the activation of self-reactive T cells to reduce autoimmunity and immunopathology [8]. Accordingly, it is suggested that the PD-1 pathway plays a crucial role in acute and chronic infections. In particular, patients infected with H. pylori express a higher level of PD-1, which leads to the exhaustion of T cells and cancer progression [9]. Moreover, human and mouse subjects presented increased PD-1 expression for 48 h after the onset of sepsis [10]. In addition, PD-1 blockade increased long-term survival in septic mice [11]. Based on these observations, it is hypothesized that a better understanding of the PD-1/PDL-1 pathway in newborn sepsis is important for its potential use as a biomarker for proper patient selection. This study aimed to determine critical parameters that are associated with the outcome of newborns with suspected sepsis.

## 2. Materials and Methods

### 2.1. Study Setting and Participant Enrollment

A prospective cohort study was performed to enroll newborns no older than 4 days who were admitted to the intensive care unit with suspected newborn sepsis in Karaganda maternity hospitals, Kazakhstan, from October 2020 to December 2021. The study was approved by the Bioethics Committee of Karaganda Medical University No. 19, dated 05 August 2019. Parent(s) of newborns gave written informed consent to participate in the study.

We excluded infants in the following circumstances: (1) newborns born of HIV-positive parents, (2) newborns treated with high-dose glucocorticoid steroid therapy, (3) newborns in a primary immunodeficiency state, (4) newborns with blood loss, or (5) refusal of the parent(s) or legal representative of the patient to participate in the study. A total of 210 newborns were included in the final analysis.

### 2.2. Clinical Data Collection

The clinical characteristics of infants were obtained, and blood samples (0.5–1 mL of venous blood) were collected in tubes with heparin at the time of taking blood cultures on the first day of hospitalization before any therapy procedures were performed in the intensive care unit.

### 2.3. Diagnostic Criteria for Clinical Sepsis and/or Culture-Proven Sepsis

The diagnostic criteria for neonatal sepsis were identified by qualified neonatologists in the intensive care unit in accordance with the “Protocol bacterial sepsis of the newborn” proposed by the Expert Commission on Health Development Ministry of Health of the Republic of Kazakhstan in 2014 [12]. According to this protocol, sepsis manifestation is based on systemic infection disease of newborns, which occurs beyond 28 days with characteristic clinical symptoms and/or positive blood culture. Neonatal sepsis patients met one or several of the following inclusion criteria: (1) high C-reactive protein (CRP) and procalcitonin and/or (2) positive bacteriological blood test, anemia, leukocytosis or leukopenia, thrombocytopenia, temperature instability, and/or organ dysfunction symptoms (central nervous, cardiovascular, and digestive systems).

### 2.4. Participant Categorization and Stratification

Initially, we categorized infants into non-septic infants and septic infants. Septic infants were compatible with the diagnostic criteria for clinical sepsis and/or culture-proven sepsis. Septic infants were further stratified into two groups— surviving group and the deceased group—according to the outcome at discharge.

### 2.5. Bacteriological Analysis

The bacteriological analysis of blood was conducted on the system BD BACTEC™ FX (Peds Plus Medium). Positive blood cultures (signs of bacterial growth) were seeded using solid nutrient media (blood agar with 5% sheep blood). Microorganism identification was made by time-of-flight mass spectrometry (Microflex-LT, Bruker Daltonics, Billerica, MA, USA). Among the identified pathogens, most were *Staphylococcus* sp. (58%), *Klebsiella* sp. (14%), or *Enterococcus* sp. (7%).

### 2.6. Immunological Analysis

A 50 µL sample of whole blood was directly mixed with a cocktail of conjugated monoclonal antibodies (Table 1) and incubated for 15 min in the dark at room temperature. Thereafter, blood was simultaneously lysed and fixed with High-Yield Lyse Fixative-Free Lysing Solution (# HYL250, Life technologies, Carlsbad, CA, USA) according to the manufacturer’s instructions. The negative control was an unstained sample. Samples were acquired on Partec CyFlow Space and analyzed by FlowJo software.

Gating strategy: CD24^+^ (neutrophils), CD14^+^ (monocytes), CD3^+^CD4^+^ (T-helper lymphocytes), CD3^+^CD8^+^ (T-cytotoxic lymphocytes), CD3^+^CD16^+^/56^+^ (natural killer cells), and CD3^-^CD19^+^ (B-lymphocytes). Activation markers: HLA-DR^+^, CD 64^+^, PDL-1 (CD 274^+^), and PD-1 (CD 279^+^).

Compensation settings were applied by using FlowMax software and Kalusa Analysis (Beckman Colter, Miami, FL, USA). The study was conducted with standardized gain settings for the entire duration of the study.

The CD64 index was calculated using the ratio of the mean fluorescent intensity (MFI) of the internal positive control (CD64^+^ neutrophils) to the MFI of CD64^−^ lymphocytes, an internal negative control, according to software gate statistics.
$$\mathrm{nCD}64\;\mathrm{index}=\frac{\mathrm{MFI}\;\mathrm{CD}64^{+}\mathrm{neu}}{\mathrm{MFI}\;\mathrm{CD}64^{-}\mathrm{lymp}}$$

The blood test was performed on a Mindray hematological analyzer. The neutrophil–lymphocyte ratio (NLR) was calculated as the percentage of neutrophils divided by the percentage of lymphocytes. The platelet–lymphocyte ratio (PLR) was defined as the ratio of the number of platelets to the number of lymphocytes (expressed as 10^9^ L^−1^).

### 2.7. Statistics

Statistical analysis was performed using the nonparametric Kruskal–Wallis test (nonparametric analog of ANOVA) (Statistica 5.5 (StatSoft, Tulsa, OK, USA)). The *p*-values are given in the tables, and *p* < 0.05 is considered statistically significant. For repeated pairwise comparisons between individual groups, the Mann–Whitney *U*-test with the Holm–Bonferroni post hoc test was used (R statistics, (Compare Groups and R-statix packages)). Categorical data were defined by the Chi-square test. The cut-off parameters were determined empirically and in line with the literature.

## 3. Results

The clinical characteristics of the three groups of infants are listed and compared in Table 2.

We first assessed the blood tests of all newborn infants (*n* = 210) (Table 3).

First, the control group differed from both septic groups in terms of hemoglobin and erythrocyte levels (deceased and surviving, respectively: hemoglobin *p* = 0.003 and = 0.006; erythrocytes *p* = 0.017 and = 0.016), but significant differences were not found between the surviving septic infants and deceased septic infants (Table 3). There was a trend towards lymphocytosis in children with a subsequent negative outcome; however, this was not reflected in the median NLR and did not reach the level of significance when using multi-group comparison tests. Table 4 presents the data of the main cell subpopulations in the studied cohort. None of the main cell subpopulations in the studied cohort was correlated with gestational age or birth body weight (Appendix A Table A1).

According to the primary immune status indicators, only the level of CD8^+^ lymphocytes significantly differed between the control group and deceased newborns (*p* = 0.002). The evaluation of the level of activation and depletion markers on immune cells in a cohort of newborns is presented in Table 5.

The most striking differences in the presented cohort were detected in functional markers of immune cells. In particular, the expression of the CD14 marker differed in all three groups; while a significant difference (*p* = 0.006) was observed between the control and both experimental groups, a difference between experimental groups was not observed (Figure 1A).

Neutrophil CD64 is a common marker of neonatal sepsis and is expressed on neutrophils during activation by infectious stimuli. According to the results, CD64 expression was significantly higher in the sepsis groups in comparison with the control group, as was expected (*p* = 0.001 and *p* = 0.009). Moreover, there was a clear tendency for a lower median in the deceased group than in the surviving group, but it was not statistically significant.

The percentage of HLA-DR-positive monocytes and lymphocytes in our cohort showed no difference. However, the median percentage of HLA-DR-positive monocytes was reduced in the deceased sepsis group but not statistically significant. The same tendency was previously described and associated with negative outcomes during neonatal sepsis.

The expression of PD-1 and PDL-1 on monocytes showed no clear differences between any groups. However, CD4^+^ T cells in the sepsis group with a negative outcome presented an increasing tendency of expression of both the marker of programmed cell death (PD-1, PDL-1) (*p* = 0.006), and while this ligand was expressed to a greater extent in the group with a negative outcome, it was not statistically significant (Figure 1B).

The clear trend was mostly detected in cytotoxic lymphocytes: the level of PDL-1 on CD8^+^ lymphocytes was more than 3 times higher in the deceased group in comparison with the other two groups (*p* = 0.005; *p* = 0.01) (Figure 1C).

## 4. Discussion

Studying the inflammatory syndrome in newborns might help to better define the immune pathophysiology of neonatal sepsis [13,14]. Sepsis is a multifactorial disease. The systemic response reflects both the pathogen and the source of infection and concomitant factors and depends on the unique immune and genetic status of the host, especially when it comes to newborns in the most vulnerable and poorly understood period of life.

This study aimed to find key parameters that affect the outcome of newborns with suspected sepsis. This might help identify unique biomarkers that may be of clinical relevance for immune monitoring, outcome prediction, and targeted therapy.

The clinical manifestation of neonatal sepsis is still complicated due to variable and non-specific symptoms, including the low sensitivity and specificity of traditional laboratory markers of inflammation. Currently, no single biomarker completely covers all criteria, leading to late diagnosis, while early treatment plays a crucial role in the outcome and prognosis [13,14,15,16].

Standard inflammatory biomarkers have not presented any impact on the outcome. This study observed that the widespread marker nCD64 [17,18] was increased in patients with confirmed sepsis and was expressed to a greater extent in deceased patients, but the changes were not significant. HLA-DR expressed on monocytes is a classic marker of sepsis in late neonatal sepsis and adults, but it does not affect the outcome of neonatal sepsis during the first days of life [19,20], which was also confirmed in this study, although it tended to decrease.

In addition, the CD14 MFI of monocytes demonstrated a significant difference in children with sepsis but did not affect the disease outcome. Monocyte CD14 (mCD14) is a cell surface glycoprotein marker found on the surface membranes of monocytes and macrophages that functions as a receptor for complexes of lipopolysaccharide (LPS) and LPS-binding proteins. It promotes the inflammatory response of the host by activating a specific pro-inflammatory signaling cascade against various infectious agents. It has been implemented as a valuable diagnostic biomarker of septicemia [21,22]. Hashem et al. [22] demonstrated that diagnostic, prognostic, and predictive results of CD14 on monocytes depended on severity. Unfortunately, this study did not confirm that result, presumably due to the age of newborns. Nevertheless, a previous study did not mention the cohort age (by default, less than 28 days old), while the studied cohort age was up to 4 days old. Moreover, the cord blood of newborns presented functional immaturity of monocytes that was characterized by the low expression of extracellular markers [22].

Immune status becomes a reflection of the systemic inflammatory response, and this study was focused on finding the source of the shifts that occur during the primary immune response. While most commonly accepted measures were highly variable and did not change significantly, there were changes in CD8^+^ T cells and a clear significant downward trend in deceased children. The level of CD8^+^ lymphocytes is a rare and variable finding among sepsis biomarkers; however, CD8^+^ T cells were previously described to play a key role in mice [23] and as a biomarker of early neonatal sepsis [24,25,26]. In particular, Carey et al. [27] reported that the number of unique T cell receptors is limited during the first trimester. Therefore, preterm neonates have a restricted distribution of the T cell repertoire with lower clonality and less richness.

Cytotoxic CD8^+^ T cells are primarily responsible for the antiviral response and intracellular pathogens, while infections in our cohort were common extracellular pathogens. However, Rudd et al. [28] demonstrated that neonatal CD8^+^ T cells are phenotypically and functionally distinct from adult CD8^+^ T cells. A shift in effector cytokine production by neonatal CD8^+^ T cells plays an innate-like role during infection, as IL-8 recruits and activates neutrophils, and γδ T cells respond with a less antigen-specific, innate phenotype than adult CD8^+^ T cells. This neonatal innate phenotype maintains an inflammatory environment that is less antigen-dependent [29]. These results indicate that CD8^+^ T cells in the neonatal stage promote protection against an extracellular bacterial pathogen and appear to act during the early innate phase of the immune response.

Immunosuppression mainly affects the regulation functional fate of lymphocytes, but not the amount. The PD-1/PDL-1 pathway is one of the key negative regulators of the immune response for immune tolerance to prevent autoimmunity and tissue damage. [20]. The role of the PD-1/PDL-1 pathway is well described in adult sepsis, while studies of neonatal sepsis have mainly been on experimental sepsis models in mice and lack clinical evidence. For instance, enhanced PD-1 and PDL-1 expression on CD4^+^T cells correlated with lethality and secondary complications [30]. The expression of PD-1 and PDL-1 on CD4^+^ and CD8^+^ T lymphocytes in patients with candidemia leads to lymphocyte exhaustion [31]. In particular, He et al. [27] showed that increased PD-1 and Tim-1 expression on CD4^+^ and CD8^+^ T cells leads to the exhaustion phenotype in mice.

The inhibitory role of the PD-1/PDL-1 pathway in the newborn was proven by secondary lung damage [32]. Furthermore, Zasada et al. [33] confirmed the trend of PD-1 expression in late neonatal sepsis and increased PD-1 expression on monocytes in terminal patients in comparison with surviving newborns. In this study, a tendency of PD1/PDL-1 expression on monocytes was determined, but the result was not significant. According to the data, the results for target cells can be correlated with the observation period [34].

Notably, the main finding of this study is the increased expression of PD-1 and PDL-1 on CD4^+^ T cells in both sepsis groups, which was not correlated with clinical outcomes, while the increased expression of PDL-1 on CD8^+^ T cells was significant between deceased and surviving newborns. The importance of this finding is associated with the possibility of using PD-1 as a therapeutic target. The blockade of the PD-1/PDL-1 pathway decreased immune dysfunction associated with sepsis in the experimental study and ex vivo. Treatment with α-PD-1 and α-PDL-1 induces bacterial clearance and mitigates organ dysfunction, recovers the immune reaction, and improves survival [35,36].

Currently, therapy based on α-PD-1 (Nivolumab) and α-PDL-1 (BMS-936559) antibodies is undergoing ongoing clinical trials to evaluate their effect on severe sepsis/septic shock [34]. However, there are no trials in children, while the theory that immunosuppression in the neonatal and adult immune systems takes place during sepsis needs to be confirmed. Moreover, this study contributes to and supports this mechanistic theory.

Furthermore, there is another way to use this biomarker. Sepsis is a heterogeneous disease predominantly characterized by hyper-inflammation or hypo-inflammation and needs better stratification and personalization [7,37]. Correspondingly, personalization of therapy with not only PD-1 and PDL-1 inhibitors but also corticosteroid therapy, which induces PD-1/PDL-1 expression [33], might improve patient outcomes. Notably, the expense of the PD-1 expression test is equal to the CD4/CD8 expression test, which is performed by clinical flow cytometry.

Taken together, the dynamic and multidirectional nature of the host immune response in sepsis might play a key role in the outcome of the disease. According to this hypothesis, it is possible to set a goal to find biomarkers for stratification. It should help identify patients at high risk of adverse outcomes, which will serve as the beginning of personalization in neonatal sepsis. One of them might be PDL-1 expression on CD8^+^ T cells, and it might be identified as a therapeutic target to fight severe inflammatory diseases, such as neonatal sepsis.

## Figures and Tables

**Figure 1 children-09-01171-f001:**
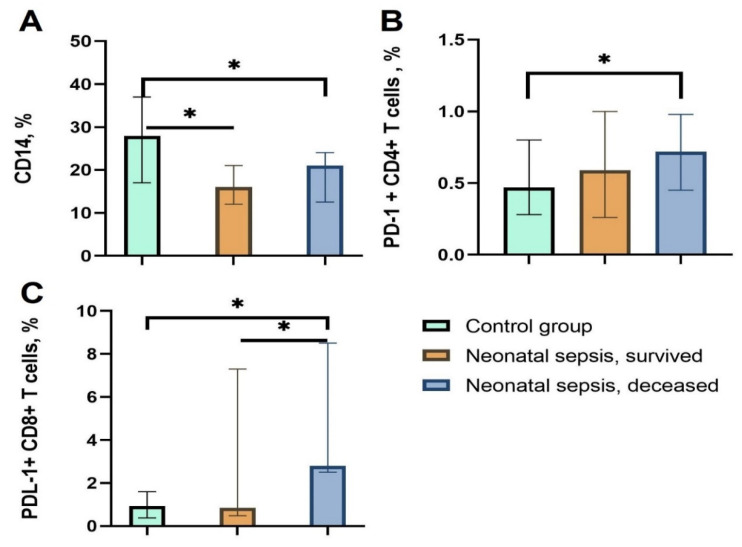
The expression level of functional markers of blood cells in newborns hospitalized with sepsis in three groups. (**A**) The percentage of expression of CD14 on monocytes; (**B**) the percentage of expression of PD-1 on CD4^+^ T cells; (**C**) the percentage of expression of PDL-1 on CD8^+^ T cells; * *p* < 0.05.

**Table 1 children-09-01171-t001:** Antibodies used throughout the research project.

Antibody	Clone ID	Source
FITC αHuman CD24	555427	BD Pharmingen
FITC αHuman CD14	555397	BD Pharmingen
Purified α-Human CD3/CD4/CD8	MA1-12474	BD Simultest
Purified α human CD3/CD16^+^CD56	342403	BD Simultest
PE α-human HLA-DR	555561	BD Pharmingen
PE α-human CD64	558592	BD Pharmingen
PE-Cy7 α-human CD19	557835	BD Pharmingen
PE-Cy7 α-human CD274	558017	BD Pharmingen
PE-Cy7 α-human CD279 (PD-1)	561272	BD Pharmingen

**Table 2 children-09-01171-t002:** Clinical characteristics between groups of the neonatal cohort.

Parameter	Non-Septic Infants *n* = 100	Surviving Septic Infants *n* = 80	Deceased Septic Infants *n* = 30	*p*-Value
Birth weight, grams,	2390 (898)	1927 (990)	1576 (860)	**0.016**
Gestational age, weeks, mean (SD)	35 (3.5)	32.2 (5.1)	29.6 (4.1)	**0.018**
Cesarean section, %	50	61	70	**0.016**
CRP, mg/L, mean (SD)	4.6 (6.5)	9.5 (11.2)	5.6 (10.1)	**0.01**
Procalcitonin, ng/mL	0.24 (3.1)	0.57 (1.6)	1.6 (5.1)	0.31
Culture-proven pathogens, Gram-positive%: Gram-negative%	-	71:29	75:25	0.54

*p*-value: Kruskal–Wallis test for comparison of 3 groups, *p* < 0.05 is significant (bold); SD: standard deviation; M: Mean, CRP: C-reactive protein.

**Table 3 children-09-01171-t003:** The main indicators of the hemogram. Venous blood was analyzed by a hematological analyzer in three groups.

Parameter	Non-Septic Infants *n* = 100	Surviving Septic Infants *n* = 80	Deceased Septic Infants *n* = 30	*p*-ANOVA
HGB, g/L, median (IQR)	174.8 (162–193)	162 (139–177)	159 (142–170)	**0.008**
RBC, ×10^9^/L, median (IQR)	4.55 (3.9–4.9)	4.1 (3.5–4.66)	3.9 (3.66–4.2)	**0.038**
Platelet, ×10^12^/L, median (IQR)	144 (106–190)	143 (80–189)	131 (80–187)	>0.05
Platelet–lymphocyte ratio, median (IQR)	2.8 (1.9–4.5)	3.2 (1.9–5.6)	2.6 (1.66–3.9)	>0.05
WBC, × 10^9^/L, median (IQR)	15.2 (11.7–22.6)	15.5 (11.5–23.7)	17.4 (8.9–33.2)	>0.05
Leukopenia, count/total amount (%)	0	3/80 (3.7%)	2/30 (6.6%)	>0.05
Leukocytosis, count/total amount (%)	27/100 (27%)	12/30 (40%)	24/80 (33%)	>0.05
Lymphocytes%, median (IQR)	48.8 (37.8–66.0)	49.7 (36.1–75.7)	68.2 (47.5–79.5)	>0.05
Neutrophils%, median (IQR)	35 (24.8–51.5)	36.3 (19.8–45.4)	24.8 (14.8–41.8)	>0.05
Lymphocytes, ×10^9^/L, median (IQR)	7.25 (5.0–10.55)	6.5 (4.4–11.7)	8.9 (4.5–15.3)	>0.05
Neutrophils 10^9^ L^−1^, median (IQR)	5.6 (3.75–7.6)	4.7 (2.6–7.3)	4.3 (1.7–6.75)	>0.05
NLR, median (IQR)	0.67 (0.36–1.25)	0.72 (0.33–1.3)	0.71 (0.27–1.1)	>0.05
NLR > 1, count/total amount (%)	34/100 (34%)	24/80 (30%)	11/30 (36%)	>0.05

*p*-value: Kruskal–Wallis test for comparison of 3 groups, *p* < 0.05 is significant (bold); IQR: interquartile range (Q1–Q3); Q1: first (lower) quartile; Q3: third (upper) quartile; HGB: hemoglobin; RBC: red blood cell; PLT: platelet; WBC: white blood cell; PLR: platelet–lymphocyte ratio; NLR: neutrophil–lymphocyte ratio. Leukopenia was defined as WBC < 5 × 10^9^/L; leukocytosis was defined as WBC > 20 × 10^9^/L.

**Table 4 children-09-01171-t004:** Immune status in newborns hospitalized with sepsis.

Parameter	Non-Septic Infants	Surviving Septic Infants	Deceased Septic Infants	*p*-Value
CD3%, median (IQR)	65.5 (57.5–71.5)	62.0 (50.0–71.0)	58.5 (48.0–72)	>0.05
CD4%, median (IQR)	45 (36.0–50.0)	39.0 (29.0–51.0)	39.5 (34.0–46.0)	>0.05
CD8%, median (IQR)	20.0 (15.0–26.0)	18.0 (14.0–250.)	15.0 (13.0–24.0)	**0.03**
CD4/CD8, median (IQR)	2.1 (1.3–2.7)	2.1 (1.4–3.1)	2.5 (1.7–3.4)	>0.05
CD19%, median (IQR)	21 (13.0–28.0)	20.6 (12.0–29)	21.9 (15.0–32.3)	>0.05
CD56/16%, median (IQR)	13.0 (8.5–17.0)	15.0 (8.0– 24.0)	13.3 (9.1–20.0)	>0.05
CD3^+^CD56/16^+^%, median (IQR)	0.64 (0.12–1.59)	0.32 (0.09–1.2)	0.7 (0.28–2.7)	>0.05

*p*-value: Kruskal–Wallis test for comparing 3 groups, *p* < 0.05 is significant (bold); IQR: interquartile range (Q1–Q3); Q1: first (lower) quartile; Q3: third (upper) quartile.

**Table 5 children-09-01171-t005:** Functional markers of blood cells in newborns hospitalized with sepsis.

Parameter	Non-Septic Infants	Surviving Septic Infants	Deceased Septic Infants	*p*-Value
CD14 MFI Mon, Median (IQR)	28 (17.0–37.0)	16.0 (12.0–21.0)	21.0 (12.5 24.0)	**0.002**
CD 64 Index, Median (IQR)	3.2 (2.2–5.2)	9.2 (2.9–17.7)	5.7 (3.22–9.0)	**0.001**
CD 64 Index > 4, count/total amount, (%)	31/100 (31%)	20/30 (66%)	53/80 (66.2%)	**0.001**
HLADR^+^ Mon%, Median (IQR)	97.5 (90–99)	95.0 (82–99.0)	90 (63.0–99.0)	>0.05
HLADR^+^ Lymp%, Median (IQR)	6.8 (3.8–10.4)	6.4 (3.4–15.3)	7.0 (4.35–16.5)	>0.05
PDL-1 (CD 274) Mon%, Median (IQR)	0.87 (1.16–2.0)	1.14 (0.44–5.2)	1.3 (0.93–5.0)	>0.05
PD-1 (CD 279) Mon%, Median (IQR)	69.0 (34.1–78)	67 (61–72)	75.5 (70.0–80.5)	>0.05
PDL-1 (CD 274) CD4%, Median (IQR)	0.47 (0.28–0.8)	0.59 (0.26–1.0)	0.72 (0.45–0.98)	>0.05
PD-1 (CD 279) CD4%, Median (IQR)	45.0 (26–65)	62 (30–73)	74.0 (68.5–79.0)	**0.012**
PDL-1 (CD 274) CD8%, Median (IQR)	0.94 (0.38–1.6)	0.85 (0.48–7.3)	2.8 (2.5–8.5)	**0.02**
PD-1 (CD 279) CD8%, Median (IQR)	83 (73–92)	79 (74.0–84.6)	85.0 (78.0–90.0)	>0.05

*p*-value: Kruskal–Wallis test for comparing 3 groups, *p* < 0.05 is significant (bold); IQR: interquartile range (Q1–Q3); Q1: first (lower) quartile; Q3: third (upper) quartile; MFI: mean fluorescent intensity; Mon: monocytes; Lymp: lymphocytes.

## Data Availability

The data presented in this study are available upon request from the respective author. The data are not publicly available until permission is obtained from the funding government agency after the official scientific report (November 2022).

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
