# Peer review of "The Presence of PDL-1 on CD8+ Lymphocytes Is Linked to Survival in Neonatal Sepsis"

_children, 2022, doi:10.3390/children9081171_

Round 1

Reviewer 1 Report

Dear Authors,

Thank you for this work and manuscript. Neonatal sepsis (both EOS and LOS) continues to be an important cause of mortality and morbidity. There are no specific biomarkers for sepsis to this day. Please see my specific comments:

Introduction:

- State in 1-2 sentences regarding what is known regarding PD-1/PDL-1 so far in pediatric/adult literature

- Aim/hypothesis should be clearly mentioned in the introduction

Materials and Methods:

- It is notable that 'MOST' of the subjects in your study were PREMATURE and particularly the 'experimental group'.

- The control group and experimental group seem quite different and thus hard to make clear comparisons. Is it possible to obtain a control group with similar gestational ages? This seems to be the biggest flaw in an otherwise will done study

- You state that the experimental group is all newborns with confirmed neonatal sepsis that should have a 'positive bacteriologic test" per the inclusion criteria. However, in Table 1. on Page 3: it does state that only 25% of the neonates w/ sepsis deceased group and 47% of the neonate w/sepsis survived group have a positive blood culture. In Page 2, Line 77: positive bacteriologic test: is that a blood culture? What test is this? Please clarify. 

Results:

It is very interesting to note that there is no significant difference in leucocyte count, lymphocytes, platelets, neutrophils or NLR in all the groups. Only the CD8 levels were different between control and sepsis/deceased group. The CD14 marker did differ significantly between control and experiment group. The PD-1 and PDL-1 on CD4+ T cells did differ in the sepsis group. Please mention these results separately. Please present these specific results in graphical format. 

- Minor comment:

In table 1. the groups are in the following order: 1.Control, 2. Neonatal sepsis, deceased and 3. Neonatal sepsis, survived.

In subsequent tables (table 3, 4, 5) - the order is reversed: 1.Control, 2. Neonatal sepsis, survived and 3. Neonatal sepsis, deceased.

Please keep this order uniform so as so not confuse the readers.

Discussion:

- Please focus your discussion on the significant findings (CD8, CD 14, PD-1 and PDL-1 in CD4 cells). Elaborate on these a little more, particularly PD-1 and PD-L-1 as potential therapeutic targets. 

- What is the turn-around time for these tests. How quickly can we get the results to make a difference in the management and outcome of these patients? This will be vital for clinicians at the bedside.

Thank you.

Author Response

 Best regards!

Reviewer 2 Report

This study prospectively included neonatal sepsis patients and compared the immunological markers with control group. The authors found PDL-1 expression on CD8+ T cells might play an immunosuppressive role during neonatal sepsis and can be used as a laboratory biomarker.

1.     In table 1, statistical comparison between deceased and survival groups needs to be performed.

2.    Neonatal sepsis could be diagnosed by clinical suspicion. However, some neonatal diseases will be presented in similar features. For example, perinatal asphyxia, RDS or patent ductus arteriosus. The major concern is some patients in neonatal sepsis group didn’t have culture-proven. The inaccurate of diagnosis will bias the study result. Also, the authors stated “positive bacteriological blood test” in inclusion criteria. Suggest clarify the inclusion criteria.

3.    The rationale about why choosing these biomarkers should be mentioned in method.

4.    All abbreviation should be spelled out below the tables. For example: Me, Mon, Lymp.

5.    Suggest English editing before publication.

Author Response

Dear Reviewer! 

Best Regards!

Round 2

Reviewer 2 Report

The authors have addressed  all the issues I concerned. I have no further comment on this manuscript.

Author Response

Dear Reviewer, thank you for your comment! 

Best regards!